# New Type of Nanocomposite CsH_2_PO_4_-UiO-66 Electrolyte with High Proton Conductivity

**DOI:** 10.3390/molecules27238387

**Published:** 2022-12-01

**Authors:** Valentina Georgievna Ponomareva, Elena Sergeevna Shutova, Konstantin Aleksandrovich Kovalenko, Vladimir Petrovich Fedin

**Affiliations:** 1Institute of Solid State Chemistry and Mechanochemistry SB RAS, Kutateladze, 18, 630128 Novosibirsk, Russia; 2Nikolaev Institute of Inorganic Chemistry, SB RAS, Ak. Lavrentiev, 3, 630090 Novosibirsk, Russia

**Keywords:** proton conductivity, cesium dihydrophosphate, UiO-66, surface interaction, nanocomposite electrolytes

## Abstract

New (1−x)CsH_2_PO_4_–xUiO-66 electrolytes with high proton conductivity and thermal stability at 230–250 °C were developed. The phase composition and proton conductivity of nanocomposites (x = 0–0.15) were investigated in detail. As shown, the UiO-66 matrix is thermally and chemically suitable for creating composites based on CsH_2_PO_4_. The CsH_2_PO_4_ crystal structure remains, but the degree of salt crystallinity changes in nanocomposites. As a result of interface interaction, dispersion, and partial salt amorphization, the proton conductivity of the composite increases by two orders of magnitude in the low-temperature range (up to 200 °C), depending on the UiO-66 fraction, and goes through a maximum. At higher temperatures, up to 250 °C, the conductivity of nanocomposites is close to the superprotonic values of the original salt at low UiO-66 values; then, it decreases linearly within one order of magnitude and drops sharply at x > 0.07. The stability of CsH_2_PO_4_-UiO-66 composites with high proton conductivity was shown. This creates prospects for their use as proton membranes in electrochemical devices.

## 1. Introduction

Over the past few decades, solid-state ordered porous materials, including metal-organic frameworks (MOFs) and covalent–organic frameworks, have attracted considerable attention due to their large and adjustable pore sizes and specific surfaces, which provide a variety of technological applications, and, in particular, the possibility of creation of proton conductors [1,2,3,4,5]. As for the creation of conductivity, there are three different ways to create proton conductors based on MOFs. The introduction of acidic groups can be carried out through the ligands, into the skeleton or the pores and channels of MOFs to realize an increase in proton concentration and mobility. The first two types of conductors based on MOFs exhibit conductivity in an atmosphere with high humidity. Frequently used acidic groups or acid molecules include, as a rule, –COOH, −SO_3_H, −PO_3_H, H_2_SO_4_, H_3_PO_4_, and ammonium cations [6,7,8,9,10]. These areas are being actively developed at the present time. Due to the complexity of the structure of MOFs, materials may exhibit different proton conductivity values and mechanisms. In [11], an approach was proposed for obtaining new proton-conducting materials by introducing proton conductors into the pores of MOFs. This approach was implemented in [12] with the introduction of triazole into MOF-containing sulfogroups in a framework and in [13] with the introduction of imidazole into a nonconducting polymer. In both cases, microporous, one-dimensional polymers with channel sizes less than 10 Å were used as a host matrix, and organic molecules with similar and predominant basic properties were used as guests. As a result, inclusion compounds are formed with an almost unchanged crystal structure compared to the host and guest molecules in the micropores, and they have a conductivity of 5 × 10^−4^ S·cm^−1^ at 150 °C [12] and 2 × 10^−5^ S·cm^−1^ at 120 °C [13At the same time, it is of significant interest to use guest compounds of other types—primarily inorganic ones with predominantly acidic properties—as proton conductors., Such compounds—in particular, the acid salts of alkali metals—have been used. Their significant advantage is the presence of superionic phases [14,15,16,17]. Other MOFs, which differ by pore size, are also of interest. For this purpose, the mesoporous Cr(III) terephthalate, also called Cr-MIL-101, with a regular 3D system of channels and a large pore volume, was used as a heterogeneous matrix for the synthesis of nanocomposites with CsHSO_4_ and CsH_5_(PO_4_)_2_ acid salts [18,19]. As a result, the nanocomposite CsHSO_4_-MIL-101 system reveals a chemical stability up to 190–200 °C and has proton conductivity that varies in a wide range: 10^−7^–10^−2^ S/cm at 50–200 °C, which exceeds the original salt by two to three orders of magnitude in the low-temperature phase. The unusual behavior of CsHSO_4_ in the nanospace of Cr-MIL-101 is associated with the appreciable interface interaction and disordered state of salt in the nanospace of mesoporous Cr-MIL-101, which causes the proton conductivity to increase [18]. An analysis of the phase composition, electrotransport characteristics, and thermodynamic properties of CsH_5_(PO_4_)_2_-Cr-MIL-101 systems with different Cr-MIL-101 contents shows that CsH_5_(PO_4_)_2_ exists in a dispersed and amorphous disordered state, which leads to an increase in proton conductivity. Due to the optimal energy of adhesion between the components, the highly dispersed and amorphous salt particles are organized in the cavities of a matrix, forming highly conductive continuous interfaces. The value of the conductivity of nanocomposites also exceeds, by more than three orders of magnitude, the ones for the original salt and dispersed matrix. As a result, nanocomposites with a proton conductivity ~10^−2^ S∙cm^−1^ at 130 °C in low humidity conditions were obtained [19]. This method, however, is challenging because introducing substances requires that the MOFs have high stabilities. The important problem is obtaining hybrid materials stable at temperatures higher than 200 °C for use as proton membranes in medium-temperature electrochemical devices [1,20]. Among the large family of MOFs, Cr- and Zr-based MOFs such as Cr-MIL-101 and UiO-66, which exhibit rich structure types, high thermal and chemical stability, and intriguing properties and functions, are regarded as some of the most promising MOF materials for practical applications [21,22]. The stability of MOFs is governed by multiple factors, including the pKa of ligands, the oxidation state, the reduction potential, the ionic radius of metal ions, the metal–ligand coordination geometry, and the hydrophobicity of the pore surface, all of which largely affect the structure of the inorganic building units and the overall binding energy between the inorganic and organic parts of the framework [23]. To date, most MOFs have exhibited poor thermal, chemical, and mechanical stability, which has limited their use in large-scale industrial applications. At the same time, the zirconium-based MOF known as microporous zirconium (IV) terephthalate ([Zr_6_O_4_(OH)_4_(bdc)_6_]·*n*H_2_O (−H_2_bdc—terephthalic acid), with the abbreviation UiO-66) has unprecedented stability and a very high specific surface area (1180–1240 m^2^/g). UiO-66 is currently one of the most stable porous carboxylate MOFs. UiO-66 is composed of Zr_6_O_4_(OH)_4_ octahedra that are 12-fold connected to adjacent octahedra through terephthalate linkers, resulting in a highly packed fcc structure [24,25] (Figure 1). The Zr-O bonds formed between the cluster and carboxylate ligands are believed to be the source of increased stability in the crystal structure of Zr-based MOFs. The exceptional mechanical stability of UiO-66 is due to its high framework connections, related to the high degree of coordination between Zr–O metal centers and the organic linkers. Specifically, the combination of strong Zr−O bonds and the ability of the inner Zr_6_-cluster to reversibly rearrange upon the removal or addition of μ_3_-OH groups without any changes in the connecting carboxylates are believed to contribute to the greater stability of UiO-66.

The chemical stability of UiO-66 is demonstrated upon washing it in boiling water and underheating it in air to 300 °C for 6 h. The UiO-66 structure has high mechanical stability and does not collapse up to pressures of 10,000 kg/cm^2^, and it retains its crystallinity even after exposure to external pressure. It decomposes above 500 °C and has good resistance to most chemicals [25,26,27]. The UiO-66 framework does not undergo hydrolysis in the pH range from 0 to 10 [25,26]. The framework of UiO-66 features two types of cages:

The cages of UiO-66 are octahedral (diameter of 9–11 Å) and tetrahedral (diameter of 7–9 Å). These cages are accessible to guest species, but they are normally filled with solvent molecules, which can be removed by heating them under a vacuum [28]. Therefore, they can be a promising and stable matrix for the introduction of salts to obtain proton membranes for medium-temperature fuel cells. Cesium dihydrogen phosphate is one of the most proton-conducting solid electrolytes of the family of acid salts of alkaline metals. The high conductivity of the superionic phase reaching ~6∙10^−2^ S/cm, and the stability of the hydrogen atmosphere makes CsH_2_PO_4_ a prospective proton membrane for various electrochemical devices, in particular, for intermediate temperature fuel cells [29,30,31,32,33,34,35,36]. Fuel cells working above 200 °C provide several advantages—such as the low sensitivity of the catalyst to CO impurity, high energy output, and efficient heat recovery [35,36]—versus polymer low-temperature fuel cells, which are very expensive, because polymer membrane materials require water management and noble metals, as catalysts for electrode reactions. Moreover, the prolonged use of low-temperature fuel cells causes the catalyst to be poisoning, which is caused by gas impurities, gradually lowering the device’s efficiency. Operations at elevated temperatures not only prevent poisoning but also significantly increase the kinetics of the electrochemical reactions, which opens the possibility of using less active but much cheaper noble-metal-free catalysts. The low-temperature conductivity of CsH_2_PO_4_ is known to be four orders of magnitude lower than that of the superionic phase, which, along with its mechanical properties, hinders its use in fuel cells. Numerous investigations have been directed for the modification of the transport properties of CsH_2_PO_4_ in the LT phase and the stabilization of the superionic phase [37,38,39]. A significant increase in LT conductivity has been obtained for nanocomposites based on various heterogeneous additives, such as SiO_2_, SiP_2_O_7_, etc., in high humidity conditions [40,41]. Dispersed additives with acidic properties and high water retention (SnP_2_O_7_, NdPO_4_) have been shown to increase the LT conductivity of CsH_2_PO_4_ and influence the stability of the HT phase in composites [39,40,41].

In this research, the zirconium-based UiO-66 was used as a matrix for the creation of nanocomposites based on CsH_2_PO_4_. This work is directed to the synthesis of (1–x)CsH_2_PO_4_-*x*UiO-66 proton electrolytes with different ratios and the investigation of their phase compositions, structural properties, and proton conductivities at various temperatures.

## 2. Results and Discussion

Microporous zirconium terephthalate UiO-66 is one of the most studied metal–organic frameworks due to its high thermal and chemical stability, as well as its good porosity [23,24,25]. Nevertheless, the characteristics of samples synthesized by different authors can vary greatly. The main factors influencing the characteristics are different synthetic conditions (for example, the modulator used or the starting reagent and their ratio) and the defective structure of the MOF obtained, which is the most pronounced in the case of UiO-66 samples. The experimental PXRD pattern of the synthesized UiO-66 at an ambient temperature coincides with data from the literature [26] and the calculated positions of Bragg reflections (Figure 2). 

It should be noted that the ideal UiO-66 composition of [Zr_6_O_4_(OH)_4_(bdc)_6_] is often unattainable due to it forming defective structures in which the modulator partially replaces terephthalate ligands, which are removed upon activation. There are two types of structural defects related to missing linker defects, in which some terephthalates are substituted by modulator carboxylic acid (formic acid in our case) and missing cluster defects, which can be considered a combination of six missing linker defects surrounded by one Zr-cluster. According to [42,43], the defects of the missing linker in UiO-66 tend to “unite” in such a way as to form a defect in the missing cluster, and, thus, additional mesopores are formed in UiO-66. It is known that increasing the amount of modulator leads to decreasing the terephthalic acid content. According to [42], the composition of UiO-66 after activation is given by the formula [Zr_6_O_6+x_(bdc)_6−x_], where x is a parameter dependent on the modulator nature and the modulator/Zr ratio. In the case of formic acid and a modulator/Zr ratio close to 100, the x parameter should be close to 1. The modulator/Zr ratio in our synthesis is ca. 88. Thus, it can be concluded that, when activated at 250 °C, the UiO-66 produced in our synthesis has the following composition: [Zr_6_O_7_(bdc)_5_]. This is confirmed by thermal analysis data. The presence of defects is associated with an increase in porosity. 

The measured isotherm of nitrogen adsorption at 77 K for UiO-66 is represented in Figure 3a. The compound under investigation possesses a type Ia isotherm according to the official IUPAC classification, which is typical for microporous compounds with narrow slit pores. 

The total pore volume is 0.61 cm^3^/g. Pore size distribution was calculated using the QSDFT equilibrium approach. Pore size distribution plots show the quite narrow pores with diameters of 6, 10, 12.5, and 15Å, which are in good agreement with the crystal structural data. The pore sizes are sufficient for the introduction and distribution of CsH_2_PO_4_ salt.

The used framework easily adsorbs water with the formation of a hydrated form of [Zr_6_O_4_(OH)_4_(bdc)_x_]·11.5H_2_O under ambient conditions. Thermogravimetric data show that when UiO-66 is heated guest water molecules are first removed from the pores (up to 100–180 °C). At higher temperatures (200–350 °C), dehydroxylation occurs with the retention of the UiO-66 crystal structure, accompanied by a change in the structure of the zirconium-containing clusters to [Zr_6_O_7_(bdc)_5_] (Figure 4). The complete decomposition of the framework begins only at 510 °C. The observed TGA curve of the UiO-66 obtained is in good agreement with [42] and reveals three steps of UiO-66 degradation: (i) up to ~180 °C—guest water molecules removed; (ii) 200–350 °C—dehydroxylation of Zr units accompanied by H_2_bdc removal and loss of modulator (iii) >500 °C—decomposition. 

The PXRD patterns of the synthesized substances are presented in Figure 5. The experimental PXRD pattern of the CsH_2_PO_4_ completely coincided with [29]. At room temperature CsH_2_PO_4_ belongs to the monoclinic singony with the P2_1_/m space group, with unit cell parameters: a = 7.9072 Å, b = 6.3869 Å, c = 4.8792 Å, β = 107.71°, and Z = 2 [29]. The superionic cubic phase of CsH_2_PO_4_ (Pm-3m) is characterized by the unit cell parameter of a=4.961Å [44]. PXRD patterns of nanocomposites at room temperature (Figure 5) show that the acid salt within retains the crystal structure of CsH_2_PO_4_ (P2_1_/m). There are no reflexes from other phases; therefore, the chemical interaction between the acid salt and the dispersed UiO-66 matrix in the composite can be excluded. 

The intensity of the most intensive peak (011) decreases more significantly than the mass fraction of CsH_2_PO_4_ in the hybrid compound. The reflexes become broader despite being tested in a humid atmosphere. For example, the intensity of the peaks is more than halved at x = 0.035, while the salt content in the composite is 81 wt%. In addition, the reflexes of CsH_2_PO_4_ (P2_1_/m) in the nanocomposites are shifted toward higher angles. The inset shows the change in intensities of the 011 and −111 reflexes in nanocomposites at x = 0.035 versus CsH_2_PO_4_ salt. It can be seen that the reflex 2θ = 23.63° is shifted to 23.70° due to a slight decrease in the unit cell parameters of the CsH_2_PO_4_ in the UiO-66 polymer matrix. The reduced intensity of the reflexes and their broadening is associated with salt dispersion and partial salt amorphization in the composite. It should be noted that the reflexes of UiO-66 are not noticeable due to the low weight fraction of MOF in the nanocomposite. 

The FTIR spectra of the CsH_2_PO_4_, UiO-66, and (1−x)CsH_2_PO_4_–xUiO-66 composites are presented in Figure 6. The most characteristic absorption bands (a.b) of UiO-66 and acid phosphate ion are observed in the range 1600–700 cm^−1^. The bands of UiO-66 correspond to data in the literature [45,46]. The absorption bands remain in the nanocomposites (marked by * in Figure 6) and become more pronounced with an increase in the mole fraction of UiO-66. The FTIR spectra and XRD data confirm the stability of the UiO-66 structure during the impregnation of the acid salt. The FTIR spectrum of CsH_2_PO_4_ is divided into two ranges: 2800–1700 cm^−1^, with intense absorption bands corresponding to the stretching and overtones of bending vibrations in the OH^−^ groups involved in hydrogen bonds (Figure 6a), and 1300–600 cm^−1^, corresponding mainly to the spectral range of the stretching vibrations of PO_4_ tetrahedra (Figure 6b). The intensive absorption bands from 2800 to 1500 cm^−1^ for CsH_2_PO_4_ are consistent with a system of strong hydrogen bonds. The a.b ν_OH_ 2684 cm^−1^ in the composite system (x = 0.018) shifts to a higher frequency, 2738 cm^−1^, and δ_OH_ 2301 cm^−1^ shifts to 2336 cm^−1^ (these bands are marked with dotted lines). There are also minor changes in the P-O stretching vibration frequencies: the absorption band νP-O 925 cm^−1^ shifts toward the higher frequency to 933 cm^−1^; a.b. at 1064 cm^−1^ shifts to 1072 cm^−1^; and a.b. 1120 cm^−1^ shifts to 1126 cm^−1^. These changes are consistent with a decrease in the length of P-O bonds and a weakening of hydrogen bonds in comparison with the pure CsH_2_PO_4_. 

The FTIR spectroscopy data for the systems based on the alkaline metal acid salts of the (M_m_H_n_(XO_4_)_p_) family and silica show that the formation of the composites occurs through the weak hydrogen bonds between salt protons and the surface OH^−^ groups of the highly dispersed silica [37,47,48]. A similar process involving the partial sorption of the protons of CsH_2_PO_4_ by the oxygen atoms of Zr clusters and the carboxylic groups of terephthalates [49,50,51,52] takes place in the CsH_2_PO_4_–UiO-66 system, which results in a weakening of the hydrogen bonds and a strengthening of the P-O bonds, leading to an increase in the mobility of protons. 

The impedance plots of the composites (x = 0.026 and 0.115) at different temperatures are presented in Figure 7. The impedance hodograph consists of an arc responsible for electrode processes (in the region of lower frequencies) and a single semicircle due to the electrolyte transfer in the higher frequencies The proton conductivity was calculated based on the values of resistance with a minimum capacitive component. With an increasing temperature, the radius of the semicircle decreases markedly. The impedances at 215 °C, in red, and at 225 °C Figure 7c,d show the significant decrease of resistance with temperature due to the phase transition of salt in the nanocomposite to the superionic phase. Thus, the proton conductivity increases with the temperature due to an increase in the number of the current carriers and their mobility.

The temperature dependencies of the proton conductivity for (1–x)CsH_2_PO_4_–xUiO-66 in comparison with the original salt are presented in Figure 8. The CsH_2_PO_4_ has a superprotonic conductivity of ~2 10^−2^ S∙cm^−1^ at 230 °C, with the activation energy of ~0.42 eV and a phase transition to a low-temperature (LT) modification involving a sharp decrease of the conductivity. Accordingly, there are two regions on the temperature dependence of the conductivity related to the superprotonic and low-temperature phases of CsH_2_PO_4_. The LT proton conductivity of CsH_2_PO_4_ obeys the Arrhenius law, with an activation energy of ~0.9 eV. The LT conductivity of CsH_2_PO_4_ is not higher than 3 × 10^−7^ S/cm at T < 200 °C due to the strong hydrogen bond network, which impedes proton transfer due to the high energy of the defect formation. Two temperature ranges can also be distinguished in the conductivity of the CsH_2_PO_4_-UiO-66. The heterogeneous doping of CsH_2_PO_4_ results in an increasing proton LT conductivity by ~1–2 orders of magnitude depending on the composition and temperature. The activation energy of proton conductivity decreases with x, from 0.9 eV for CsH_2_PO_4_ to 0.62 eV for x = 0.018 and 0.77 eV for x = 0.035 and 0.07. A decrease in the activation energy with an increase in the inert additive, is typical for the solid composite electrolytes [47,48] and is characterized by partial salt amorphization and the contribution of the interfaces to the overall conductivity, the number of which increases with x. A high value of LT conductivity in the composite is most likely associated with a high degree of binding between the acid protons of CsH_2_PO_4_ and oxygen in the UiO-66 matrix while maintaining the composition and crystal structure of the salt, as was observed for the other nanocomposites [37,47,48]. This provides the high adhesion values and as a result an increase in the conductivity. CsH_2_PO_4_ can completely fill the nanopores of the UiO-66, which are accessible to salt, as well as the intergranular space at low x values, forming continuous proton-conducting pathways. Figure 9 shows the isotherms of conductivity as a function of the volume and mole fraction of UiO-66 for a range of temperatures. LT conductivity increases two orders of magnitude, up to x = 0.035, and then decreases slowly. Up to x ~0.035, the volume fraction and pore volume of UiO-66 are less than the volume of the salt fraction, and the conductivity passes through a gentle maximum, which comes close to filling the pores and is equal to 25–50 vol.% of UiO-66 (Table 1, Figure 9). With higher UiO-66 content (x > 0.07), the pore volume markedly exceeds the total volume occupied by the salt, and the conductivity decreases due to the “conductor–insulator” percolation effect. Similar dependencies with maximum conductivity were observed for CsHSO_4_-Cr-MIL-101 and CsH_5_(PO_4_)_2_-Cr-MIL101 systems at 55–65 vol.% [18,19]. It is very important that the conductivity of nanocomposites in the high-temperature phase is close to that of the CsH_2_PO_4_ at a low UiO-66 fraction, up to x = 0.018 (~23 vol.% UiO-66), and it reaches of 1.3 × 10^−2^ S/cm at 235^o^C at p_H2O_ = 0.3 atm. Then, superionic conductivity decreases linearly up to x = 0.07 and drops sharply with the dispersed matrix content due to the percolation effect “conductor-isolator”. 

It should be noted that the conductivity values are reproducible in repeated heating-cooling cycles at 50–250 °C and are stable during long-term storage at high temperatures (Figure 10). 

The isothermal exposure of the sample at 235 °C and 0.3 atm H_2_O did not lead to a decrease in conductivity with time. The values of the proton conductivity remained unchanged over the course of three hours. This potentially shows the stability of the composite (1−x)CsH_2_PO_4_−xUiO-66 solid-state system at high temperatures. Thus, highly conductive nanocomposites based on a metal–organic framework, stable at temperatures of up to 250 °C, were obtained for the first time.

## 3. Materials and Methods

Crystals of CsH_2_PO_4_ were grown via isothermal evaporation from aqueous solutions of phosphoric acid (analytical purity grade) and cesium carbonate (high purity grade) taken in the reguired molar ratio at room temperature. XRD patterns were fully consistent with the Inorganic Crystal Structure Database for CsH_2_PO_4_ (P2_1_/m) at room temperature. UiO-66 was synthesized according to [1] using a slightly modified synthetic procedure. In a typical synthesis, 27 mL of DMF were mixed with 13 mL of 85% HCOOH (ca. 0.29 mol) and heated up to 100 °C. Then, ZrO(NO_3_)_2_·2H_2_O (0.89 g; 0.0033 mol) was added to the resulting mixture. After obtaining a clear solution, terephthalic acid was added (0.553 g; 0.0033 mol) in small portions with vigorous stirring for 10 min. The mixture was heated up to 130 °C and kept at this temperature for two days with constant stirring. The resulting white precipitate was filtered off using a Buchner funnel and then washed in a filter with hot DMF (2 × 0.5 mL), hot water (4 × 0.5 mL), and acetone (3 × 0.5 mL). The crude product obtained was further dried in an oven at 80 °C overnight and then activated at 250 °C for 3 h; the yield was 0.96 g. Its structural characteristics also corresponded to UiO-66. The porous structure of UiO-66 was analyzed using the nitrogen adsorption technique on a Quantochrome’s AutosorbiQ gas sorption analyzer at 77 K. Initially, the compound was activated under a dynamic vacuum at 250 °C for 18 h to remove water molecules from the compound. The nitrogen adsorption−desorption isotherms were measured within the range of relative pressures, from 10^−6^ to 0.995. The specific surface area was calculated from the data obtained using the conventional BET, Langmuir, and DFT models. According to the nitrogen heat adsorption method, the synthesized UiO-66 had a specific surface area of 1520 m^2^/g and a pore volume of ~0.61 cm^3^/g. Pore size distribution was calculated using the DFT method.

The (1–x)CsH_2_PO_4_-xUiO-66 electrolytes (molar ratio 0 < x < 0.15) were prepared via the thorough multiple mixing of the components in an agate mortar. The molar fractions were selected following various degrees of pore-filling, including the complete filling of UiO-66 pores and an excess or lack of salt. The volume fractions of the UiO-66 matrix for these compositions are presented in Table 1. 

The transition from the hydrated form into the dehydrated UiO-66 took place in a temperature range of 100–350 °C. The powders of CsH_2_PO_4_ and dehydrated UiO-66 were mixed thoroughly in an agate mortar and then compacted at 300–400 MPa, together with two platinum electrodes, into pellets of 0.1–0.15 cm in thickness and 0.4–0.6 cm in diameter. The tablets were then heated at a temperature of 230–240 °C at 0.3 atm H_2_O to form composites.

The conductivity (σ) of the samples was measured by electrochemical impedance spectroscopy in a two-electrode cell with a Precision LCR Meter IPU-1RLC-1/2008 in an ac-frequency range of 1–3.3 MHz and Instek (the frequency range of 12 Hz–200 kHz). The conductivity measurements were carried out in a cooling regime (~0.5–1 degree/min) in humid conditions at 180–250 °C to prevent CsH_2_PO_4_ dehydration. The required water vapor pressure (P_H2O_~0.3 atm) was maintained by flowing argon at a fixed rate (30 mL/min) through the bubbler with water at a temperature of 70 °C. The conductivity measurements at temperatures below 180 °C were carried out in an argon atmosphere or in air with a relative humidity of 15-20%. When measuring conductivity, the samples were repeatedly heated and cooled at a temperature range of 50–250 °C. The conductivity (σ) was measured using analysis of impedance plots at each temperature and was calculated according to the following equation: σ = l R^−1^ S^−1^, where l is the thickness, R is the electrolyte resistance, and S is the effective electrolyte area. The relative error of determining the conductivity was less than 1–3%. Thermal gravimetric analyses were carried out using a NETZSCH TG 209 F1 Iris Thermo Microbalance at 25–600 °C in helium flow with a heating rate of 10 °C·min^−1^. The phase composition of nanocomposites was analyzed using a D8 Advance Diffractometer (λCuK_α_1 = 1.5406 Å) with a one-dimensional Lynx-Eye detector and a K_β_ filter at room temperature. The initial structure information was taken from the Inorganic Crystal Structure Database (FIZ Karlsruhe, Germany) and Cambridge Structural Database accessed on 2008. (https://www.ccdc.cam.ac.uk). The FTIR spectra in the attenuated total reflectance (ATR) mode were recorded with a Bruker Tensor 27 spectrometer.

## 4. Conclusions

In this study, we developed an approach related to the introduction of proton centers into the pores of metal–organic frameworks. Nanocomposite electrolytes based on MOF and superionic salt CsH_2_PO_4_ were proposed and synthesized for the first time. As a result, (1−x)CsH_2_PO_4_−xUiO-66 proton conductors of different compositions (x = 0–0.15) with high conductivity and stability in a range of medium temperatures (up to 260 °C) were studied in detail. The thermally and chemically stable UiO-66 was shown to be a suitable matrix for the introduction of CsH_2_PO_4_. The optimal ratio of components in nanocomposite with the high proton conductivity has been determined. Two temperature ranges can be distinguished in the conductivity of CsH_2_PO_4_-UiO-66 composites related to the high superionic and low-temperature phases of CsH_2_PO_4_. It was shown that a CsH_2_PO_4_ crystal structure remains in nanocomposite systems, but dispersion and partial salt amorphization were observed due to the interface’s interaction with a microporous UiO-66 matrix. As a result, the conductivity in the low-temperature region increased up to two orders of magnitude, going through a smooth maximum (25–50 vol.% of UiO-66), and it decreased due to the percolation “conductor–isolator” effect. The maximum LT conductivity values (10^−5^ S/cm for f_vUiO-66_ = 40% at 160 °C) corresponded to the complete filling of pores. The superprotonic conductivity of CsH_2_PO_4_ in the nanocomposite with low UiO-66 content (f_vUiO-66_ = 9%) was close to that of the original salt (1.3 × 10^−2^ S/cm at 235 °C at p_H2O_ =0.3 atm). This value is one of the highest and most significant values for conductors based on MOFs. The time stability of the high conductivity values of the (1−x)CsH_2_PO_4_-xUiO-66 nanocomposites was shown at 240–250 °C. This creates prospects for their use as medium-temperature proton membranes based on CsH_2_PO_4_. The systems under study exhibited features characteristic of previously studied composites based on acid salts and dispersed oxides. A more detailed study of interfacial interaction and the mechanism of proton conductivity are of great interest.

## Figures and Tables

**Figure 1 molecules-27-08387-f001:**
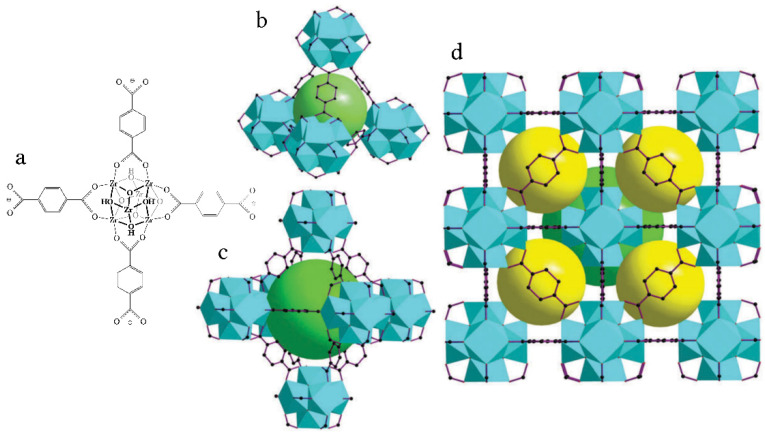
Illustration of the crystalline structure of UiO-66. (**a**) Structure of the inorganic building unit {Zr_6_O_4_(OH)_4_(O_2_CR)_12_}: Zr atoms are located at the vertices of octahedra, which are included in the cubes of the μ_3_-O and μ_3_-OH ligands; every two Zr atoms are connected by carboxylate ligands (only 4 of them are shown for clarity) forming 12 connected nodes; these nodes are combined, forming tetrahedral (**b**) and octahedral (**c**) cavities; (**d**) packing of two types of cavities according to [26]; tetrahedral—yellow; octahedral—green.

**Figure 2 molecules-27-08387-f002:**
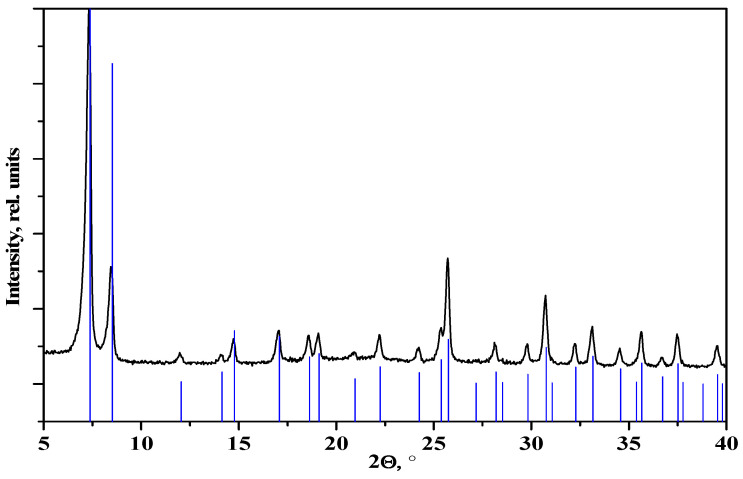
PXRD pattern of synthesized UiO-66; vertical markers—calculated positions of Bragg reflections of UiO-66.

**Figure 3 molecules-27-08387-f003:**
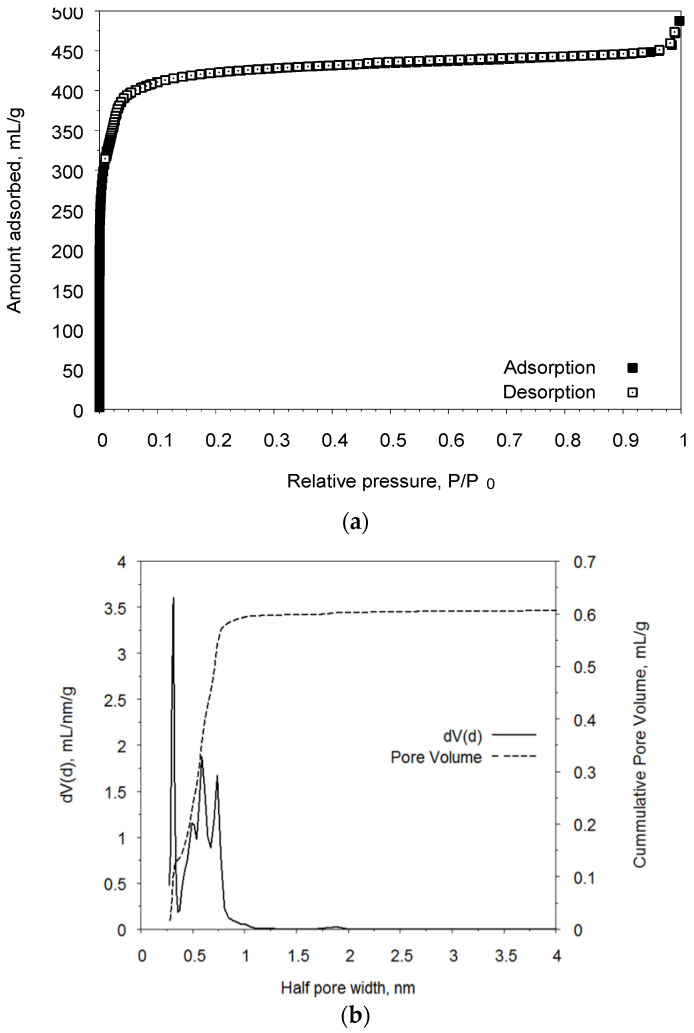
(**a**) Isotherm of the adsorption of N_2_ at 77 K for the sample UiO-66. (**b**) Pore size distribution calculated using the DFT method.

**Figure 4 molecules-27-08387-f004:**
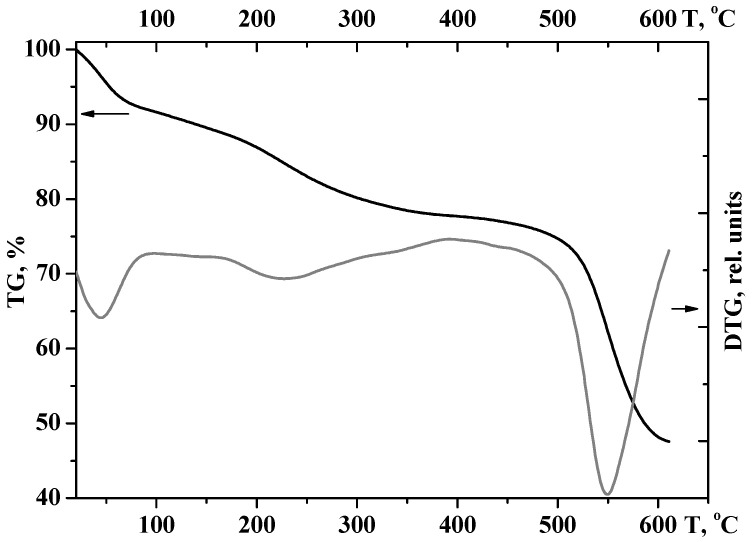
Thermogravimetric data for UiO-66, as synthesized (heating rate of 10 °C·min^−1^).

**Figure 5 molecules-27-08387-f005:**
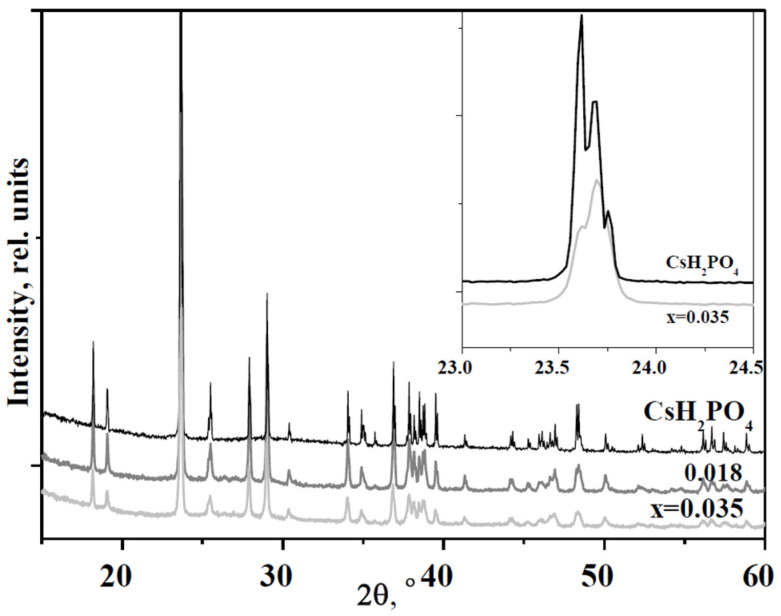
PXRD patterns of (1−x)CsH_2_PO_4_-xUiO-66 in comparison with CsH_2_PO_4_. The inset shows the 011 and −111 reflexes.

**Figure 6 molecules-27-08387-f006:**
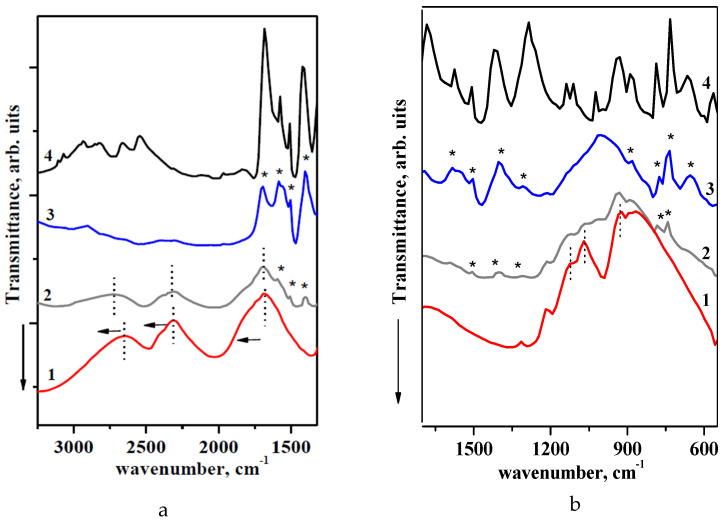
FTIR spectra of (1−x)CsH_2_PO_4_ -xUiO-66 for different compositions: 1—CsH_2_PO_4_; 2—x = 0.018; 3—0.035; 4—UiO-66.

**Figure 7 molecules-27-08387-f007:**
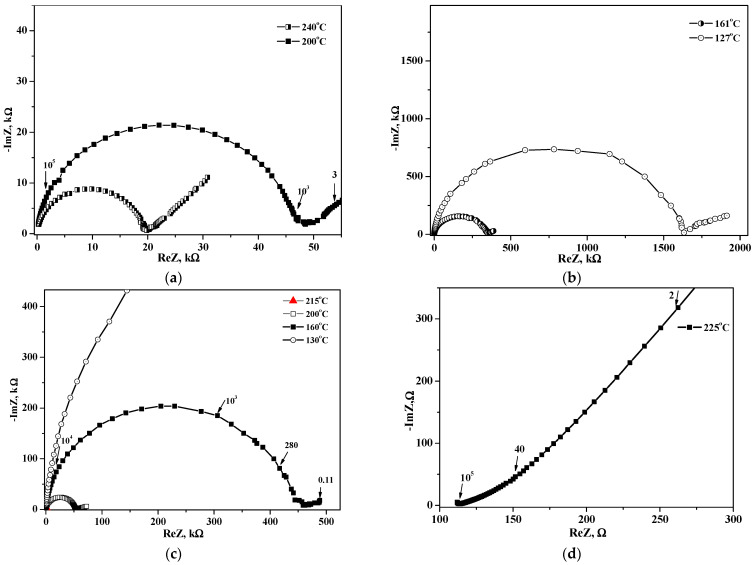
Electrochemical impedance spectra of (1–x)CsH_2_PO_4_–xUiO-66 as a function of temperatures for x = 0.115 (**a**,**b**) and 0.026 (**c**,**d**); the numbers indicate the current measurement frequencies in Hz.

**Figure 8 molecules-27-08387-f008:**
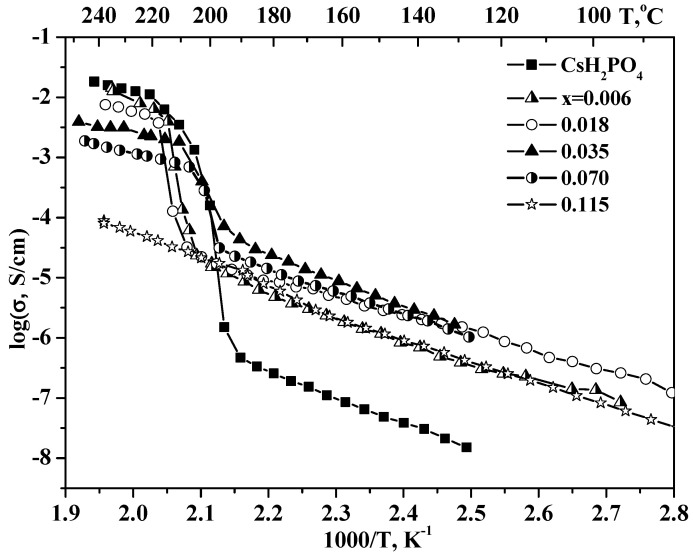
Temperature dependences of conductivity for initial CsH_2_PO_4_ and (1−x)CsH_2_PO_4_-xUiO-66 of different compositions; x = 0.006–0.115 (cooling regime, 0.5–1 deg/min, P_H2O_~0.3 atm at 180–250 °C, and in air with RH=15–20% at 50–180 °C).

**Figure 9 molecules-27-08387-f009:**
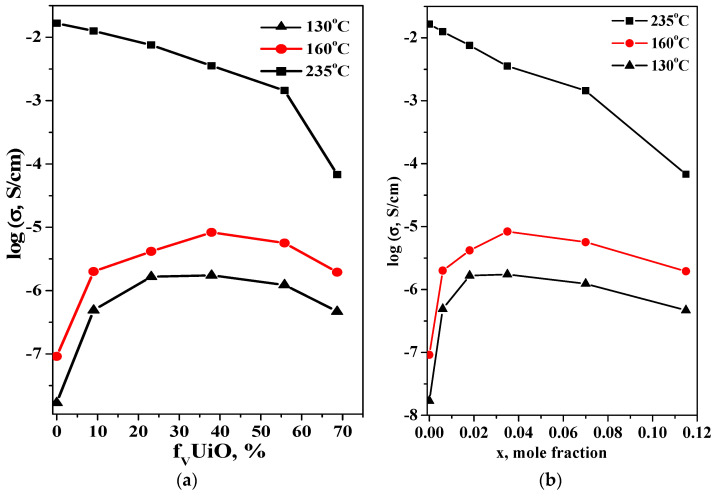
The isotherms of conductivity for CsH_2_PO_4_-UiO-66 as a function of the volume (**a**) and mole (**b**) fraction of UiO-66.

**Figure 10 molecules-27-08387-f010:**
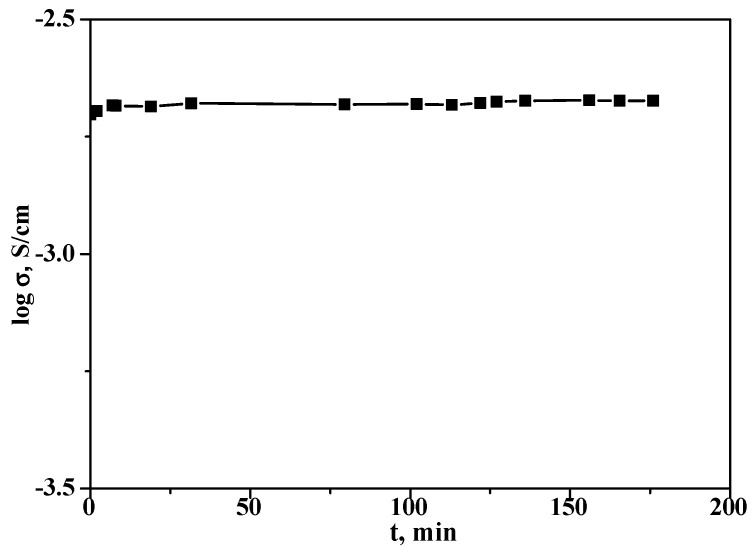
The time dependence of proton conductivity for (1−x)CsH_2_PO_4_−xUiO-66 (x = 0.035) (isothermal holding: 240 °C, pH_2_O = 0.3 atm).

**Table 1 molecules-27-08387-t001:** The ratio of the volume and mole fractions of xUiO-66 in the (1−x)CsH_2_PO_4_−xUiO-66 systems.

x, mole fraction of UiO-66	0	0.006	0.018	0.026	0.035	0.070	0.0115
f, volume fraction of UiO-66, %	0	9	23	31	38	56	69

## Data Availability

Not applicable.

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
