# Peer review of "New Type of Nanocomposite CsH2PO4-UiO-66 Electrolyte with High Proton Conductivity"

_molecules, 2022, doi:10.3390/molecules27238387_

Round 1
Reviewer 1 Report
New (1-x)СsH2PO4–xUiO-66 electrolytes with high proton conductivity and thermal stability at 230-250°C were developed. Proton conductivity of composite increases by two orders of magnitude in the low-temperature range (up to 200°C), depending on the UiO-66 fraction, and goes through maximum. Long-term stability of СsH2PO4-UiO-66 composites with high proton conductivity was shown.
The article is devoted to interesting topic, but contains many problems. Therefore, it cannot be published in itscurrent form.
1. Conductivity was measured during cooling. The relative humidity was maintained by passing an argon current through water at 70°C. Obviously, the relative humidity during the cooling process should increase significantly and CsH2PO4 could partially dissolve. This can make a significant contribution to the proton conductivity of composites and distort data at low temperatures.
2. “A similar process with partial sorption of protons of CsH2PO4 by oxygen atoms of Zr clasters and carboxylic groups of terephthalates [xxxv,xxxvi,xxxvii,xxxviii] takes place in the CsH2PO4–UiO-66 systems, which results in a weakening of hydrogen bonds and strengthening of P-O bonds and leads to increase in the mobility of protons.” In the presented IR spectra, there are no bands of stretching vibrations of water, which makes it impossible to estimate the strength of hydrogen bonds.
3. “The LT conductivity of CsH2PO4 is not higher than 3*10−7 S/cm at Т < 200 °Ð¡ due to the strong hydrogen bond network which impedes proton transfer due to the 283 high energy of defect formation”. Since the composite is formed by hydrogen bonds, they must be even stronger in it. This contradicts what is written. But hydrogen bonds must be characterized.
4. “Isothermal holding of the sample at 235°C and 0.3 atm H2O does not lead to a decrease in conductivity with time. The values of proton conductivity remain unchanged during three hours. It shows the long-term stability of conductivity for (1-x)CsH2PO4-xUiO-66 solid state systems at high temperatures.” The figure shows that the conductivity of the material increases with time. Time is too short. It is incorrect to say that this shows the long-term stability of the material.
5. The list of references looks very archaic. It contains only 6 references to articles of the last five years, of which 2 are articles by authors. It seems that for solid proton-conducting electrolytes based on MOFs, this is completely incorrect. I recommend the authors to discuss recent papers on the proton conductivity of MOFs.
Smaller notes:
· Why are the data in Figure 8 given in the volume fraction? What is the meaning of table 1?
· “Proton conductivity of composite increases by two orders of magnitude in the low-temperature range (up to 200°C), depending on the UiO-66 fraction, and goes through maximum. Superprotonic conductivity of nanocomposites at low UiO-66 values is closed to original salt, then decreases linearly within one order of magnitude, and drops sharply at x>0.07”. It should be noted that we are talking about different temperature ranges. It is seems like inconsistency.
· The text of the abstract reproduces the conclusion verbatim in many ways.
· References in the text are given in different styles.
· In figure 6, the compositions are not indicated.
· “To the resulting mixture ZrO(NO3)2·2H2O (0.89 g; 0.0033 mol) was 131 added”. The sentence must be revised.
Author Response
Dear Referee
We are grateful for the valuable additional questions and comments.
We made the substantial changes of the text of the manuscript, taking into account all the comments.
They are highlighted in the manuscript.
The additional corrections made it possible to improve the content of the article.
New (1-x)СsH2PO4–xUiO-66 electrolytes with high proton conductivity and thermal stability at 230-250°C were developed. Proton conductivity of composite increases by two orders of magnitude in the low-temperature range (up to 200°C), depending on the UiO-66 fraction, and goes through maximum. Long-term stability of СsH2PO4-UiO-66 composites with high proton conductivity was shown.
The article is devoted to interesting topic, but contains many problems. Therefore, it cannot be published in its current form.
We absolutely agree with all the comments made by the reviewers.
- Conductivity was measured during cooling. The relative humidity was maintained by passing an argon current through water at 70°C. Obviously, the relative humidity during the cooling process should increase significantly and CsH2PO4 could partially dissolve. This can make a significant contribution to the proton conductivity of composites and distort data at low temperatures.
Page 6 shows the measurement conditions as “The conductivity measurements were carried out in the cooling regime (~1 degree/min) in humid conditions at 180oC-250oC to prevent CsH2PO4 dehydration and in air with relative humidity of 20% at temperatures 50°Ð¡-180 °Ð¡”. The same information is available in the captions to Fig. 9. Thus, at temperatures below 180°C, the passage of gas through the bubbler was stopped and the measurements were carried out in air with a relative humidity of 15–20%. The conductivity data for the original salt were obtained in the same mode and completely coincide with the literature values. We prefer to make the conductivity measurements in the cooling regime rather than the heating mode to avoid errors due to water sorption in air during the sample preparation.
- “A similar process with partial sorption of protons of CsH2PO4 by oxygen atoms of Zr clasters and carboxylic groups of terephthalates [xxxv,xxxvi,xxxvii,xxxviii] takes place in the CsH2PO4–UiO-66 systems, which results in a weakening of hydrogen bonds and strengthening of P-O bonds and leads to increase in the mobility of protons.”In the presented IR spectra, there are no bands of stretching vibrations of water, which makes it impossible to estimate the strength of hydrogen bonds.
We added the data of FTIR in the range of absorption bands, corresponding to stretching and overtones of bending vibrations of OH− groups involved in hydrogen bonds (Fig. 6(a)) and made a brief description in the text (page 11). We made an assumption in the manuscript about the mechanism of composite formation by analogy with the other systems, since the insignificant changes in the νOH and δOH and νP-O absorption bands of the CsH2PO4 salt were recorded. To establish the mechanism for the formation of the composite, it is also necessary to determine in details the changes in absorption bands of UiO-66 using the compositions with a high content of the matrix.
- “The LT conductivity of CsH2PO4 is not higher than 3*10−7 S/cm atТ< 200 °Ð¡ due to the strong hydrogen bond network which impedes proton transfer due to the high energy of defect formation”. Since the composite is formed by hydrogen bonds, they must be even stronger in it. This contradicts what is written. But hydrogen bonds must be characterized.
As a result of interface interaction of salt protons with the matrix the energy of hydrogen bond network of CsH2PO4 salt in composite weakens.
- “Isothermal holding of the sample at 235°C and 0.3 atm H2O does not lead to a decrease in conductivity with time. The values of proton conductivity remain unchanged during three hours. It shows the long-term stability of conductivity for (1-x)CsH2PO4-xUiO-66 solid state systems at high temperatures.”The figure shows that the conductivity of the material increases with time. Time is too short. It is incorrect to say that this shows the long-term stability of the material.
You are absolutely right. This is the first data obtained for the composite system. An exposure of the sample during three hours showed that the conductivity does not decrease with time, potentially showing the stability of the system. In the future, special longer tests should be carried out. Corresponding changes have been made in the text of the article.
- The list of references looks very archaic. It contains only 6 references to articles of the last five years, of which 2 are articles by authors. It seems that for solid proton-conducting electrolytes based on MOFs, this is completely incorrect. I recommend the authors to discuss recent papers on the proton conductivity of MOFs.
Thank you for your valuable comments. We added the recent papers on the proton conductivity of MOFs (16 references) and some discussion. They are highlighted in the text.
Smaller notes:
Why are the data in Figure 8 given in the volume fraction? What is the meaning of table 1?
The volume ratios between salt and a dispersed component or its pore volume are used to understand the percolation limit ”conductor-isolator” between ideally distributed phases in composite electrolytes. If the volume of the dispersed component (or pores) significantly exceeds the volume of the introduced salt, then the conductive paths are disturbed, and the conductivity usually decreases sharply. Therefore, the appropriate molar ratios were initially chosen to meet these requirements.
“Proton conductivity of composite increases by two orders of magnitude in the low-temperature range (up to 200°C), depending on the UiO-66 fraction, and goes through maximum. Superprotonic conductivity of nanocomposites at low UiO-66 values is closed to original salt, then decreases linearly within one order of magnitude, and drops sharply at x>0.07”. It should be noted that we are talking about different temperature ranges. It is seems like inconsistency.
Changes have been made to the text for greater clarity and understanding, page 11 and Conclusions.
The text of the abstract reproduces the conclusion verbatim in many ways.
Thank you for your comment. We have made a number of changes in the text of the conclusion, which is highlighted in the article.
References in the text are given in different styles.
This is our omission. We tried to make everything in the same style.
In figure 6, the compositions are not indicated.
The compositions are shown in Figure 6.
“To the resulting mixture ZrO(NO3)2·2H2O (0.89 g; 0.0033 mol) was 131 added”. The sentence must be revised.
The sentence was revised.
In addition, Figure 7 for impedance measurements and their description has been introduced.
Reviewer 2 Report
Review
The work by V.G. Ponomarevaet al. reports a study of nanocomposite CsH2PO4-UiO-66. The authors clearly define the research problem, they clearly show the importance of this study and the results would be worthy for publication in Molecules. The manuscript presents the new interesting information about the new composite. The proposed method for creating composites will probably create new ways to obtain proton-conducting systems with improved properties.
Before this article can be considered for publication, the authors should make some revision.
Some remarks and questions:
Ð .7, line 253. Could the authors explain why the decrease of the unit cell parameters of CsH2PO4 took place?
P. 8. Figure 6. IR spectra…. Explanations in the caption to the figure, what is 1,2, etc. need to be explained.
P.8. The authors talk about the “weakening of hydrogen bonds and strengthening of P-O bonds” in connection with the received IR data. The authors should explain which bands were shifted and where to confirm this statement.
The authors used electrochemical impedance spectroscopy, so the hodographs should be shown to understand what contributions the authors are analyzing when discussing electrical conductivity.
Author Response
Dear Referee
We are grateful for the valuable additional questions and comments.
We made the substantial changes of the text of the manuscript, taking into account all the comments.
They are highlighted in the manuscript.
The additional corrections made it possible to improve the content of the article.
Ð .7, line 253. Could the authors explain why the decrease of the unit cell parameters of CsH2PO4 took place?
Substances enclosed in micro- and mesopores may change their properties due to the size effect, and therefore the unit cell parameter of CsH2PO4 in composites decreases slightly.
- 8. Figure 6.IR spectra…. Explanations in the caption to the figure, what is 1,2, etc. need to be explained.
Clarifications were made in the captions to the Figure 6. Besides we added the data of FTIR by the range of absorption bands, corresponding to stretching and overtones of bending vibrations of OH− groups involved in hydrogen bonds (Fig. 6(a)) and made a brief description in the text (page 11).
P.8. The authors talk about the “weakening of hydrogen bonds and strengthening of P-O bonds” in connection with the received IR data. The authors should explain which bands were shifted and where to confirm this statement.
We made a brief description in the text. The additions are highlighted in the article, page 11.
The authors used electrochemical impedance spectroscopy, so the hodographs should be shown to understand what contributions the authors are analyzing when discussing electrical conductivity.
Figure 7 for impedance measurements and their description has been introduced (page 11, 12)
Besides, we have made a number of changes in the text of the conclusion, which is highlighted in the article. The list of references is supplemented by 16 articles.
Round 2
Reviewer 1 Report
With the described approach, when measuring conductivity, relative humidity (equal to 20% of the saturated vapor pressure of water at a given temperature, cannot be maintained in any way). You just need to put a description of the conditions without specifying the humidity.
The driving force of the composite formation should be the formation of strong bonds of salt and filler. And this bond can only be hydrogen and should not be weaker than the original one. Your claim about the weakening of bonds in the composite looks unfounded.
Sorry, but if you need volume fractions, bring them throughout the text from the beginning. Table 1 looks strange for a scientific paper.
Author Response
Dear Referee
We are grateful for the valuable additional questions and comments.
We made the substantial changes of the text of the manuscript, taking into account all the comments.
They are highlighted in the manuscript.
The additional corrections made it possible to improve the content of the article.
With the described approach, when measuring electrical conductivity, relative humidity (equal to 20% of the saturated vapor pressure of water at a given temperature, does not support in any way). You just need to put a description of the conditions without indicating humidity.
Argon flow through the water bubbler was stopped at 180°C. The cell was purged with argon. Further conductivity measurements at temperatures below 180°C were carried out in argon atmosphere or in air with a relative humidity of 15-20%. The changes are marked in the text.
The driving force behind the formation of the composite should be the formation of strong bonds of salt and filler. And this bond can only be hydrogen and should not be weaker than the original one. Your claim about the weakening of bonds in the composite looks unfounded.
We fully agree with you that the driving force in the formation of a composite is the formation of hydrogen bonds between the salt and the matrix. In the article, we tried to compare the change of absorption bands of CsH2PO4 in the composites in comparison with the original salt. The hydrogen bond network and P-O bonds change due to the binding of a proton to the O containing groups of the matrix.
Sorry, but if you need volume fractions, bring them throughout the text from the very beginning. Table 1 looks strange for a scientific article.
The volume fractions or pore volumes of matrix are important in the understanding the conductivity data. Since the molar masses of the salt and the matrix are very different, we believe that the data in Table 1 are needed to make the results easier to understand. We moved the Table to the beginning of the text of the manuscript. Fig. 9 was supplemented with by Fig. 9(b) with a dependence of conductivity on the mole fractions.
Besides, the changes in reference numbering made on page 15 References.
